# Short-Term Exposure Effects of the Environmental Endocrine Disruptor Benzo(a)Pyrene on Thyroid Axis Function in Zebrafish

**DOI:** 10.3390/ijms23105833

**Published:** 2022-05-23

**Authors:** Giuditta Rurale, Ilaria Gentile, Camilla Carbonero, Luca Persani, Federica Marelli

**Affiliations:** 1Lab of Endocrine and Metabolic Research, IRCCS Istituto Auxologico Italiano, 20100 Milan, Italy; giudi.rurale@gmail.com; 2Department of Medical Biotechnology and Translational Medicine, University of Milan, 20100 Milan, Italy; ilaria.gentile@unimi.it (I.G.); camilla.carbonero@studenti.unimi.it (C.C.)

**Keywords:** zebrafish, hypothalamic–pituitary–thyroid axis, endocrine disruptor chemicals, benzo(a)pyrene, central hypothyroidism

## Abstract

Benzo(a)Pyrene (BaP) is one of the most widespread polycyclic aromatic hydrocarbons (PAHs) with endocrine disrupting properties and carcinogenic effects. In the present study, we tested the effect of BaP on thyroid development and function, using zebrafish as a model system. Zebrafish embryos were treated with 50 nM BaP from 2.5 to 72 h post fertilization (hpf) and compared to 1.2% DMSO controls. The expression profiles of markers of thyroid primordium specification, thyroid hormone (TH) synthesis, hypothalamus-pituitary-thyroid (HPT) axis, TH transport and metabolism, and TH action were analyzed in pools of treated and control embryos at different developmental stages. BaP treatment did not affect early markers of thyroid differentiation but resulted in a significant decrease of markers of TH synthesis (*tg* and *nis*) likely secondary to defective expression of the central stimulatory hormones of thyroid axis (*trh*, *tshba*) and of TH metabolism (*dio2*). Consequently, immunofluorescence of BaP treated larvae showed a low number of follicles immunoreactive to T4. In conclusion, our results revealed that the short-term exposure to BaP significantly affects thyroid function in zebrafish, but the primary toxic effects would be exerted at the hypothalamic-pituitary level thus creating a model of central hypothyroidism.

## 1. Introduction

Thyroid follicles synthesize and secrete thyroid hormones (TH) through the activation of the hypothalamic–pituitary–thyroid (HPT) axis. Briefly, the hypothalamus secretes the thyroid-releasing hormone (TRH) that induces the pituitary gland to release the thyroid-stimulating hormone (TSH) which promotes the synthesis of TH, thyroxine (T4), and triiodothyronine (T3). TH are essential for regulating development, growth, morphogenesis, basal metabolism, reproduction, and behavior in vertebrate species [1]. There is a large body of evidence accumulated in the last two decades between environmental exposure to a specific class of chemicals and its effects on the endocrine system. The exposure to environmental toxins termed endocrine disrupting chemicals (EDCs), including polychlorinated biphenyls, dioxins, phthalates, and polycyclic aromatic hydrocarbons, has been shown to interfere with production, release, transport, metabolism, binding, and action of hormones necessary for the regulation of developmental processes maintenance of homeostasis [2,3]. EDCs have potential thyroid disruption effects, but the in vivo studies are scarce. Toxicity on the HPT axis may be caused by several mechanisms, and each EDCs may affect the thyroid system at different levels counteracting the expression of TSH, the activity of the sodium-symporter (NIS) and thyroperoxidase (TPO) enzyme, or the peripheral metabolism and action of TH operated by the deiodinase (DIO1-3) enzymes and the TH receptors (THRA and THRB), respectively [2,3,4,5,6].

Polycyclic aromatic hydrocarbons (PAHs) are a large group of diverse organic compounds that contain two or more fused aromatic rings, many of which are classified as organic pollutants with toxic and carcinogenic effect, constituting a significant hazard for human health [7,8]. In fact, PAHs are associated with an increased risk of skin, lung, bladder, and gastrointestinal cancers, likely causing DNA damage and insertion of deleterious mutations [7,9]. PAHs are introduced into the environment via natural emission (e.g., forest-prairie fires and volcanic eruption), but largely arise from human activities, such as vehicle emissions, oil shipping, and refineries [7]. Since PAHs are semi-volatile compounds, they are present in both gas and particulate phases under ambient conditions, leading to different biologically effective pathways through inhalation, ingestion, or dermal contact [10]. Despite improved legislation and monitoring, PAHs are still found in concentrations above the limits of the law in many countries, and it is fundamental to establish their sources and potential toxic effects. Currently, 16 PAHs have been designated as priority contaminants and confirmed as EDCs by the US Environmental Protection Agency (US-EPA) (archive.epa.gov, accessed on 20 May 2022). In Southern Italy, a case study reported a potential contamination by PAHs of the metropolitan areas wherein an increased lifetime cancer risk exists [11,12]. In Chinese surface water, PAH pollution is a serious problem because it causes reproductive damages to marine organisms [13]. Animal studies have shown that exposure to PAHs mixture increased plasma vitellogenin levels and disrupted gametogenesis in Atlantic cod, and induced lipid peroxidation and reproductive toxicity in sea turtles and fish [14,15,16,17]. Fish can biotransform PAHs into reactive metabolites with high reactivity for the DNA causing genotoxic effects or inducing oxidative stress through the production of reactive oxygen species (ROS) [18]. Regarding the thyroid function, epidemiological studies have associated PAHs exposure to decreased levels of circulating and tissue TH, which seems to be caused by a direct toxicity on thyroid gland, or by interfering with DIO2 synthesis or TH binding protein expression [2,4,5].

The present study is focused on benzo(a)pyrene (BaP), one of the most widespread PAHs and usually used as marker of PAH contamination in the atmosphere and marine environments [2,19]. BaP was mostly studied due to its carcinogenic properties and reproductive toxicity on marine species, but its effects on thyroid function are not well documented. *Liza abu* fish (mullet) injected with BaP showed a significant decrease in TH plasma level and increased TSH concentration, and pathological alterations of thyroid follicles [20], but the mechanism underlying thyroid toxicity is far from clarified.

In this work, we used the zebrafish model to exploit the effects of BaP on thyroid system, including early specification of thyroid primordium, TH synthesis, HPT axis regulation, and TH metabolism and action.

## 2. Results

Zebrafish embryos exposed to high BaP dosage (50 nM) developed normally up to 3 days post fertilization (dpf), where the survival rates were equivalent to that of controls treated with 1.2% DMSO. Interestingly, most of the survived BaP treated embryos died between 3 and 5 dpf, a developmental window characterized by molecular and morphological changes required for larval transition (Appendix A).

Regarding the analysis of thyroid development and function, stable transcription of the internal control was vital for the validity of the qRT-PCR examination. Among the house keeping genes, beta-actin proved to be the most suitable as internal reference. The transcriptional expression of genes responsible for early thyroid specification (*nkx2.4b*, *pax2a* and *hhex*), TH synthesis (*tg*, *nis* and *tpo*), transport (*mct8*), metabolism (*dio2* and *dio3*), and action (*thraa* and *thrb*) were investigated in zebrafish embryos after the BaP treatment at different developmental stages (Figure 1A).

At 28 hpf, no significant differences in the expression of *nkx2.4b*, *pax2a*, and *hhex* were detected in BaP treated embryos compared to controls (Figure 1A). Consistently, WISH of *nkx2.4b*, *pax2a* at the level of thyroid primordium appeared unaffected in the BaP treated embryos (Appendix A). The whole data suggested that the early events responsible for thyroid primordium specification were not disturbed by BaP exposure.

Concerning TH synthesis (Figure 1B), the expression of *tg* was only slightly reduced in BaP treated embryos at 54 hpf, while *nis* and *tpo* were significantly diminished and increased, respectively. At 80 and 120 hpf both *tg* and *nis* were significantly reduced in BaP treated embryos, where the *tpo* levels were mildly reduced. Moreover, BaP exposure showed to have a strong impact on the HPT axis. At 54, 80, and 120 hpf, both hypothalamic and pituitary markers (*trh* and *tshba*) were significantly lower in BaP-treated embryos. The *tshr* was decreased in the BaP embryos at 80 hpf, whereas at 54 and 120 hpf the mRNA levels appeared only slightly affected (Figure 1C). *Mct8* and *dio2*, which are involved in TH transport and T4 to T3 conversion, were reduced at 54 hpf. At 80 and 120 hpf, only *dio2* continued to be significantly compromised in the BaP treated embryos, whereas *dio3*, responsible for TH inactivation, was expressed at levels similar to those of controls (Figure 1D). Finally, our data indicate that BaP exposure did not alter the expression of genes involved in the TH action. In fact, both *thraa* and *thrb* expression was not significantly changed in the BaP treated embryos at all developmental stages (Figure 1E).

To confirm the qRT-PCR results we performed WISH on the BaP treated embryos at 54 and 80 hpf (Figure 2). In the vast majority of BaP treated embryos (>90%), the signals corresponding to the thyroid markers *tg* and *nis* were strongly reduced compared to what observed in controls at 54 and 80 hpf (Figure 2A–D, A’–D’). The hypothalamic *trh* and the pituitary *tshba* transcripts were barely detectable or absent in the 95% of the embryos treated with BaP at both stages (Figure 2E’–H’), confirming the central regulation of the thyroid axis as the most relevant site of the BaP effect. In addition, the BaP treatment caused a diminished expression of the *dio2* in the 98% of the embryos at the pituitary level at 54 and 80 hpf (Figure 2I’,J’). Immune staining with an anti-T4 antibody was performed to assess the status of thyroid function at 120 hpf by comparing the control and BaP treated larvae (Figure 2K–K’). By counting individual T4-positive follicles, we observed that BaP treatment was associated to a significant reduction of follicles that actively synthetize T4 (DMSO = 5.4 ± 0.81 vs. BaP = 1.6 ± 0.74) (Figure 2L).

## 3. Discussion

The present study is the first to address thyroid function after exposure to endocrine disruptor BaP dissolved in the harvesting water, using zebrafish as a model system. The findings are consistent with a disruption of TRH-TSH signal, thus creating a model of central hypothyroidism [21].

EDCs can interfere with thyroid function at multiple levels, falling into two macro categories: (i) TH synthesis or (ii) TH action disruptors [22]. However, the effects of BaP of thyroid system are far from clarified. For that reason, we have taken advantage of a zebrafish model to increase knowledge about the impact of BaP. Zebrafish embryos have proven to be a suitable model to test the effects of endocrine disruptors on thyroid development, since the surrounding morphological and molecular events are well characterized. Thyroid organogenesis starts around the 20 hpf with the expression of the early thyroid transcription factors (TFTs), *nkx2.4b*, *pax2a*, and *hhex*, forming the so-called thyroid primordium, differentiated from the 42 hpf expressing the thyroid functional markers *tg*, *nis*, *tpo* and other factors necessary for TH synthesis. From the 55 hpf the mature thyroid follicles start to proliferate and migrate along the ventral aorta and produce T4 under the stimulation operated by TSH signal [23,24,25]. We show that early specification of thyroid primordium is not affected, since the expression of the TFTs *nkx2.4b*, *pax2a*, and *hhex* at 28 hpf were preserved. However, in later stages, the thyroid markers (*tg* and *nis*), responsible for proper TH synthesis, appeared strongly reduced at the level of the thyroid tissue of BaP treated embryos. Interestingly, the expression profile of HPT axis revealed that both *trh* and *tshba* are reduced by BaP exposure, as confirmed by qPCR and WISH experiments. Additionally, the expression of *dio2* enzyme, necessary for the T3 activation, is strongly reduced in the pituitary of BaP treated embryos. The abovementioned abnormalities of TH synthesis resulted in <80% of T4 production by BaP treated larvae.

It has been shown that the early thyroid specification is independent from HPT axis regulation, since the *trh*, *tshba*, *tshr*, and *dio2* start to be expressed from 2 days post fertilization [26,27]. Opitz et al. also demonstrated that the knock-down of *tshr* affected the expression of functional thyroid markers *tg*, *nis*, and *tpo* at 55 and 100 hpf, resulting in a reduced number of follicles immunoreactive for T4 at larval stage [25]. In light of these data, since our treated embryos presented defects only in the later phases of thyroid development associated with hypothalamic and pituitary alterations, we can hypothesize that BaP did not affect thyroid organogenesis itself but is primarily acting at the hypothalamic-pituitary level generating a form of central hypothyroidism [21]. Therefore, BaP is a novel chemical to be added to the quite long list of disruptors known to interfere at various levels of TH synthesis, likely altering HPT axis functions [3]. Discrepant results on BaP toxicity obtained in a mullet fish model [20] are likely due to differences in the route of administration of the disruptor (BaP injected vs. dissolved in the harvesting water).

Many EDCs, including bisphenol A and pesticides, are reported to act as TH agonist or antagonist and impact mRNA expression of TRs in vitro [22]. Our zebrafish model displays normal levels of both *thraa* and *thrb*, suggesting that TH action is not affected by this chemical. Moreover, additional BaP effects on TR binding or expression may be masked by the maternal TH supply stored into the yolk sac and reabsorbed by the embryo during the first days of development [28]. Thus, studies on juvenile or even adult zebrafish may better clarify the role of BaP on TH action.

Regarding the effects of BaP on TH synthesis, the aberrant hypothalamic and pituitary signals make us think about a possible toxic effect on the central nervous system. Neurotoxicity has been largely overlooked regarding the effects of BaP. In the adult zebrafish brain, BaP alters locomotor and cognitive ability due to a decreased level of several neurotransmitters, and neurodegeneration associated to accumulation of amyloid b protein and cell apoptosis [29]. Previous studies performed on zebrafish embryos and larvae also described an alteration of global and gene-specific DNA methylation patterns after short-term exposure to BaP [30,31]. Moreover, acute exposure to BaP in blood clam (*Tegillarca granosa*) led to neurotoxicity, which has been associated to changes in DNA methylation, affecting the oxidative stress response and inducing cell apoptosis [32].

## 4. Materials and Methods

Zebrafish line and maintenance. All experiments were performed according to EU regulations on laboratory animals (Directive 2010/63/EU). Wild-type adults (AB strain) were maintained in a flow-through system at a constant temperature (28 °C–1 °C), with a photoperiod (light:dark) of 14:10. Zebrafish embryos were obtained from natural spawning and raised until the desired developmental stages according to established morphological criteria [33,34]. Starting from 24 h post fertilization (hpf), embryos were harvested in fish water containing 0.002% of 1-phenyl-2-thiourea (PTU; Sigma-Aldrich, St. Louis, MO, USA) to prevent pigmentation and 0.01% methylene blue to prevent fungal growth. Such a low dose of PTU (0.002%) does not affect TH synthesis [35].

BaP treatment. For each experiment, pools of 100 zygotes were raised in glass petri dish (diam. 150 mm; Merck KGaA, Darmstadt, Germany) and treated with 100 mL of BaP solution at a concentration of 50 nM in fish water (Sigma-Aldrich, St. Louis, MO, USA) prepared freshly from a stock solution of 1 mg/mL Treatments were performed from 2.5 to 72 hpf and the BaP solutions were renewed twice a day. After 72 hpf, embryos were repeatedly washed and transferred in clean fish water until 120 hpf. As control, embryos were treated with 1.2% of dimethyl- sulfoxide (DMSO, Sigma-Aldrich, St. Louis, MO, USA), used as solvent for BaP solution.

RNA extraction and quantitative PCR (qPCR). Total RNA was extracted from pools of 40–50 embryos/stage treated with BaP or DMSO at 28, 54, 80, and 120 hpf using TRIzol Reagent (Thermo Fisher Scientific, Waltham, MA, USA). cDNA synthesis reaction was carried out following the protocol of GoScript Reverse Transcription System (Promega, Madison, WI, USA). Quantitative real-time PCR (qPCR) was performed by ABI PRISM 7900HT Fast Real-Time PCR System using SYBR Green Master Mix (Thermo Fisher Scientific, Waltham, MA, USA) and the previous reported primers [27]. Beta-actin gene was used as endogenous control. Experiments were performed in triplicate and results are expressed by mean ± SD. Statistical significance was calculated using Student’s *t*-test (* *p* < 0.05 ** *p* < 0.01 *** *p* < 0.001).

Whole mount in situ hybridization (WISH) and immunofluorescence (IF). WISH experiments were performed according to Thisse and Thisse protocol [35], using DIG-riboprobes of thyroid (*tg* and *nis*), HPT axis (*trh* and *tshba*), and TH metabolism (*dio2*) markers [23,27,36]. Anti-DIG alkaline phosphatase (AP) and nitro-blue tetrazolium/5- bromo-4-chloro-3′-indolyphosphate (NBT/BCIP) were used to detect the probes in DMSO and BaP treated embryos at different developmental stages. After WISH experiments, the embryos were post-fixed in 4% PFA and analyzed in glycerol 85% under a stereomicroscope. WISH images were acquired in 90 controls and 90 BaP treated embryos derived from three independent experiments. Changes in transcript expression after BaP exposure were qualitatively evaluated counting the embryos with normal (compared to DMSO controls) or reduced/absent WISH signal.

The analysis of TH production was performed by IF using a rabbit anti-T4 BSA serum (1:1000; MP Biochemicals, Santa Ana, CA, USA) and the AlexaFluor 555 anti-rabbit IgG as secondary antibody (1:500, Thermo Fisher Scientific, Waltham, MA, USA) [37]. The embryos were rinsed in glycerol 85% and 30 controls and 30 BaP treated larvae heads were mounted in a glass slide and acquired under a fluorescent stereomicroscope. The number of T4-positive follicles were manually counted, and the results were expressed as mean ± SD. Statistical significance was calculated by Student’s *t*-test (*** *p* < 0.001).

## 5. Conclusions

Collectively the phenotype observed in our embryos leads us to propose a form of central hypothyroidism for BaP toxicity. The compromised hypothalamus-pituitary function would then hamper the adequate functional maturation of thyroid follicles and TH synthesis. Future studies will be needed to delineate the biological mechanism through which BaP affects HPT axis function.

## Figures and Tables

**Figure 1 ijms-23-05833-f001:**
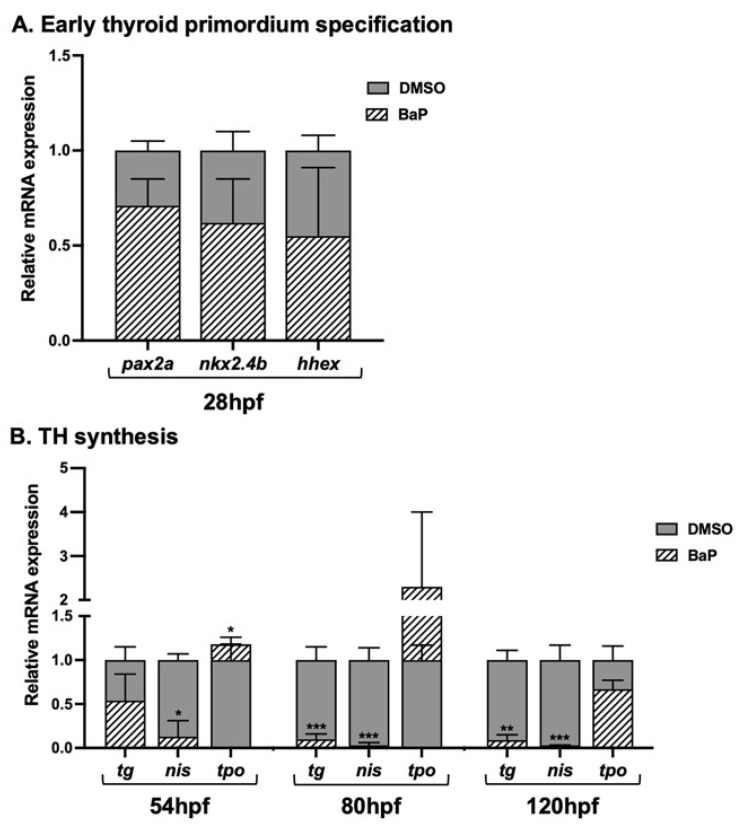
Transcriptome analysis of thyroid system. Relative mRNA expression of genes involved in (**A**) Early thyroid primordium specification: *nkx2.4b*, *pax2a*, and *hhex*; (**B**) TH synthesis: *tg*, *nis*, and *tpo*; (**C**) HPT axis regulation: *trh*, *tsh*, and *tshr*; (**D**) TH transport and metabolism: *mct8*, *dio2*, and *dio3*; (**E**) TH action: *thraa* and *thrb*. Experiments were performed in triplicate using pools of 1.2% DMSO controls and 50nM BaP treated embryos, at 28, 54, 80, and 120 hpf. Results are expressed by Mean ± SD. Statistical significance was calculated using Student’s *t*-test (* *p* < 0.05, ** *p* < 0.01, *** *p* < 0.001).

**Figure 2 ijms-23-05833-f002:**
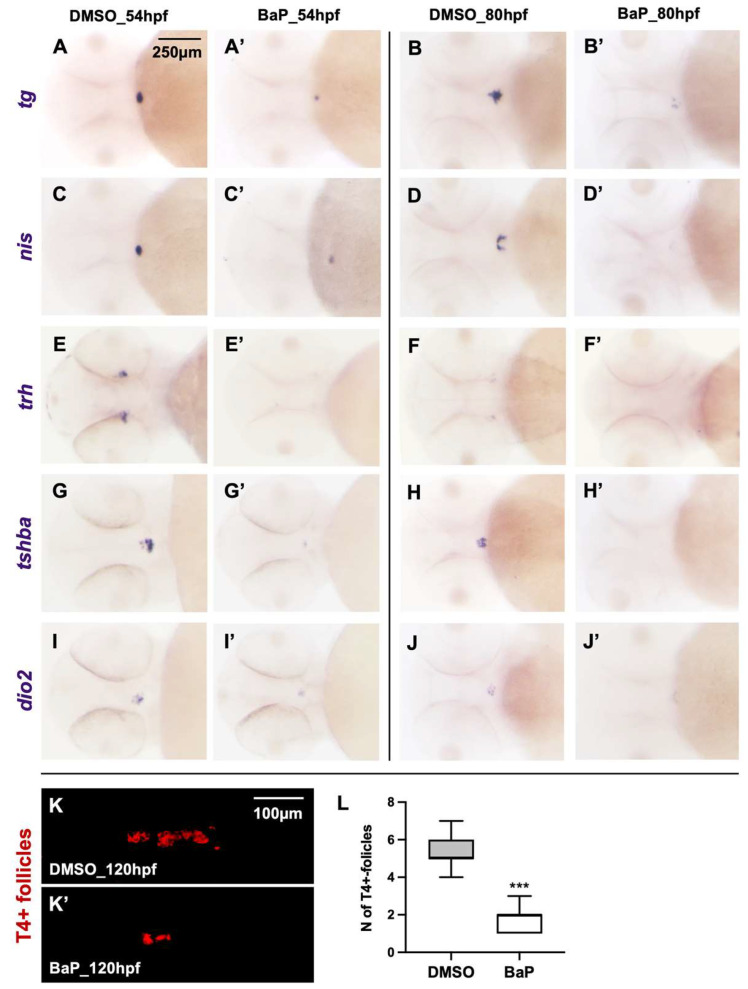
WISH and IF of thyroid function. (**A**–**J’**) WISH of HPT axis markers in BaP treated embryos (50 nM) and controls (1.2 % DMSO) at 54 and 80 hpf. (**A**–**B’** and **C**–**D’**) *tg* and *nis* at the level of thyroid tissue; (**E**–**F’**) *trh* expression in the hypothalamic region; (**G**–**H’** and **I**–**J’**) *tshba* and *dio2* in the anterior pituitary. Each experiment was performed in triplicate using about 30 embryos/stage. Embryos were acquired in dorsal (**A**–**D’**) or ventral (**E**–**J’**) views, anterior to the left. (**K**,**K’**) IF of T4 produced in thyroid follicles of DMSO and BaP treated larvae at 120 hpf. Thyroid regions were acquired mounting the heads in ventral view, anterior to the left. (**L**) Number of T4-positive follicles. Results are expressed by mean ± SD (*** *p* < 0.001).

## Data Availability

Not applicable.

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
