# Peer review of "Short-Term Exposure Effects of the Environmental Endocrine Disruptor Benzo(a)Pyrene on Thyroid Axis Function in Zebrafish"

_ijms, 2022, doi:10.3390/ijms23105833_

Round 1

Reviewer 1 Report

This is a Communication in which the investigators have evaluated the effects of exposure of zebrafish to the BaP endocrine disruptor. The effects on thyroid specification, TH synthesis, HPT axis, and TH transport and metabolism have been evaluated based on qRT-PCR gene expression for selected regulatory genes and whole mount in situ hybridization at three hpf. No change in thyroid specification in treated fish compared to control-treated fish was taken to mean proper thyroid specification of thyroid primordium. Points of concern of the study are:

  1. Why were thyroid specification data only given for 28 hpf when the study included three additional hpf? Also, at that point, the treated fish had a lower levels of gene expression for thyroid specification. The statement on line 91 needs to be amended to read “no significant differences” rather than “no differences”.
  2. The unchanged effect on TH action needs to be discussed more thoroughly in light of the study overall.
  3. The method of WISH quantification need better explanation.
  4. Why are the WISH data only shown for 54 and 80 hpf but not 120?

Author Response

We thank the reviewer for the comments.

Please see the attachment for the point-by-point response.

Reviewer 2 Report

Dear authors!

Please, add section "Conclusion" to your manuscript.

Author Response

We thank the reviewer for the positive comments. 

As requested, we added the conclusion session to the revised manuscript and we update the discussion accordingly.